# Pose Prediction of Autonomous Full Tracked Vehicle Based on 3D Sensor

**DOI:** 10.3390/s19235120

**Published:** 2019-11-22

**Authors:** Tao Ni, Wenhang Li, Hongyan Zhang, Haojie Yang, Zhifei Kong

**Affiliations:** 1School of Mechanical and Aerospace Engineering, Jilin University, Changchun 130022, China; nitao@jlu.edu.cn (T.N.); whli19@mails.jlu.edu.cn (W.L.); kongzf17@mails.jlu.edu.cn (Z.K.); 2School of Mechanical Engineering, Yanshan University, Qinhuangdao 066004, China; goodluckko@foxmail.com

**Keywords:** autonomous vehicle, LiDAR point cloud, Kalman filter, vehicle dynamics, active suspension system

## Abstract

Autonomous vehicles can obtain real-time road information using 3D sensors. With road information, vehicles avoid obstacles through real-time path planning to improve their safety and stability. However, most of the research on driverless vehicles have been carried out on urban even driveways, with little consideration of uneven terrain. For an autonomous full tracked vehicle (FTV), the uneven terrain has a great impact on the stability and safety. In this paper, we proposed a method to predict the pose of the FTV based on accurate road elevation information obtained by 3D sensors. If we could predict the pose of the FTV traveling on uneven terrain, we would not only control the active suspension system but also change the driving trajectory to improve the safety and stability. In the first, 3D laser scanners were used to get real-time cloud data points of the terrain for extracting the elevation information of the terrain. Inertial measurement units (IMUs) and GPS are essential to get accurate attitude angle and position information. Then, the dynamics model of the FTV was established to calculate the vehicle’s pose. Finally, the Kalman filter was used to improve the accuracy of the predicted pose. Compared to the traditional method of driverless vehicles, the proposed approach was more suitable for autonomous FTV. The real-world experimental result demonstrated the accuracy and effectiveness of our approach.

## 1. Introduction

Thanks to the efforts of many researchers, driverless technology has developed rapidly in the 21st century. Several research teams developed advanced autonomous vehicles to traverse complex terrain in the 2004 and 2005 DARPA grand challenges [1], and then urban roads in the 2007 DARPA urban challenge (DUC) [2]. Research related to self-driving has continued at a fast pace not only in the academic field but also in the industrial field. Some driverless taxis from Google and driverless trucks from TuSimple have entered the stage of commercial operation. Localization and perception are two significant issues relating to driverless technology. Accurate localization can ensure the safety of autonomous vehicles on the road without collision with surrounding objects. Multi-sensor fusion is a stable and effective method for locating autonomous vehicles on the road [3,4,5,6], which can achieve centimeter-level localization accuracy by fusing data from GNSS, LiDAR, and inertial measurement unit (IMU). The perception element of vehicles is mainly completed by cameras equipped to the inside of the car. Onboard computers use some algorithms to detect, classify, and identify the information captured by the camera. The deep neural network is an effective method for recognizing the information obtained from the camera [7,8,9]. Although researchers have done a lot of research on driverless cars, most of the research on autonomous vehicles has been carried out on an urban even road, rarely considering uneven terrain. However, uneven terrain can seriously affect the stability of autonomous full tracked vehicle (FTV), especially some vehicle-mounted equipment, such as laser lidar and inertial navigation system (INS). How to maintain the stability and safety of the vehicle on uneven terrain is an important issue. One way to solve the problem is to apply an active suspension system to the vehicle, which calculates the road inputs of vehicles in advance through preview information [10,11,12,13,14,15,16].

In the last two decades, 3D laser scanners were widely used in autonomous vehicles because it could provide high-precision information about the surrounding environment [17,18,19,20]. Several methods to combine 3D sensors with preview control have been proposed. Youn et al. [21] investigated the stochastic estimation of a look-ahead sensor scheme using the optimal preview control for an active suspension system of a full tracked vehicle (FTV). In this scheme, the estimated road disturbance input at the front wheels was utilized as preview information for the control of subsequently following wheels of FTV. Christoph Göhrle [22] built a model predictive controller incorporating the nonlinear constraints of the damper characteristic. The approximate linear constraints were obtained by a prediction of passive vehicle behavior over the preview horizon using a linearized model. The simulation result showed improved ride comfort.

Accurate pose estimation can be used not only as an input parameter of the active suspension system but also as a reference for path planning. Yingchong Ma [23] presented a method for pose estimation of off-road vehicles moving over uneven terrain. With the cloud data points of terrain, accurate pose estimations can be calculated used for motion planning and stability analyses. Julian Jordan [24] described a method for pose estimation of four-wheeled vehicles, which utilized the fixed resolution of digital elevation maps to generate a detailed vehicle model. The result showed that the method was fast enough for real-time operation. Jae-Yun Jun [25] proposed a novel path-planning algorithm as a tracked mobile robot to traverse uneven terrains, which could efficiently search for stability sub-optimal paths. The method demonstrated that the proposed algorithm could be advantageous over its counterparts in various aspects of the planning performance.

Many filtering techniques have been widely used in engineering practice. Kalman filter is a kind of filtering technology, which is mainly used to correct errors caused by model inaccuracies. Dingxuan Zhao [26] proposed an approach using the 3D sensor, IMU, and GPS to get accurate cloud data points of the road. Both GPS/INS loosely-coupled integrated navigation and Kalman filter (KF) were used to get accurate attitude angle and position information. The results demonstrated that the KF could effectively improve the performance of the loosely coupled INS/GPS integration. Hyunhak Cho [27] presented an autonomous guided vehicle (AGV) with simultaneous localization and map building (SLAM) based on a matching method and extended Kalman filter SLAM. The proposed method was more efficient than the typical methods used in the comparison.

In these theses of pose prediction, vehicles were generally regarded as a rigid body, which was not in line with the characteristics of the vehicle’s wheels. Moreover, most of the research on preview control did not give a detailed description of adjusting the suspension system through terrain information. Furthermore, there are few studies on autonomous FTV. In this paper, we proposed an approach to predict the pose of autonomous FTV using GPS, IMU, and 3D laser scanner. In Section 2, we introduced the configuration of the autonomous FTV and the overall flow of our approach. In Section 3, the dynamic model of autonomous FTV was established. Then, we presented the process of combining a Kalman filter with a dynamic model of the vehicle and the control method of the active suspension system. Finally, we demonstrated the effectiveness and accuracy of our approach with real-world experiments.

## 2. System Structure

In the driverless field, how to ensure the safety and stability of vehicles on uneven roads is a significant issue. Changing the control strategy by getting road information and calculating its impact on vehicles in advance is an almost perfect way to solve the issue. Three-dimensional laser scanners, IMU, and GPS are necessary to implement the above method. Our vehicle was equipped with an IMU, two GPS, two 3D laser scanners, and an active suspension system, as shown in Figure 1. With 3D laser scanners, we could obtain real-time points cloud data around the vehicle for extracting elevation information of the road. The IMU and GPS were used to get accurate attitude angle and position information of the vehicle. Then, the contact height between wheels and the ground could be obtained by integrating the 3D laser scanner, IMU, and GPS data. Ultimately, the vehicle’s pose could be calculated by importing the wheels’ height information into the dynamic model. However, the predictive pose was inaccurate due to the error caused by sensors. To improve the precision of the pose, a Kalman filter was used to compensate for the errors caused by sensors. Figure 2 shows the main steps.

### Mapping and Location

In the last two decades, sensor technology that can obtain information about the surrounding environment has been rapidly developed, such as depth camera, lidar, and vision sensors. With real-time environmental information, autonomous vehicles can achieve safe autonomous navigation in a complex environment. In this paper, we selected 3D laser scanners to obtain road information because of their many advantages, such as high precision of measurement and good stability in complex environments.

Two Velodyne lidars were chosen as the 3D sensor of our autonomous vehicle in our research, just as shown in Figure 1. The 3D sensor has a sweep angle of 360° and a horizontal angle resolution of 0.1°–0.4°, which ensures to obtain high-density points cloud data. The lidar has many kinds of frequencies. The scanning frequency of 20 Hz meets the need for extracting the height information of the vehicle’s wheels. Furthermore, a lidar was installed at the bottom of the vehicle in our study, which means that it needed a larger measuring range. Velodyne lidar could meet our needs. Table 1 shows the main performance index of the lidar.

Location is significant because it is a prerequisite for us to extract the height information of the vehicle’s wheels. Inertial measurement units are used to measure the physical information of the vehicle. For example, we could calculate the position, velocity, and attitude angle of the vehicle with the outputs of IMU. However, the IMU has accumulated errors because it obtains location information of vehicles in the form of mathematical integration. The global position system is widely used in the field of autonomous vehicles because of its advantage of providing a long-term stable location in all weathers. Although GPS has a higher positioning accuracy and better stability than other positioning methods, it will cease to be effective in some scenarios. For example, GPS signals will be interfered with by high buildings in cities and shielded in tunnels, which causes the car to be unable to locate itself. Some studies have been carried out on the integration of IMU and GPS data for positioning. Our GPS/INS system was based on differential positioning and RTK technology, ensuring centimeter-level positioning accuracy. Table 2 shows the performance index of our system.

## 3. Method

### 3.1. Vehicle Dynamics

Our autonomous FTV mainly adopted the multi-axle steering method to improve its flexibility, ensuring that it could choose more control strategies for avoiding obstacles on the terrain. The front axle and rear axle were steering axles, which were used to change the vehicle’s direction, and the middle shaft is the driven shaft. Figure 3 shows the simplified model of the vehicle.

The steering angle of each wheel could be calculated as follows:(1)when turning right, θ>0
(1)$$\left\{ \begin{array}{l} \theta_{1}=\theta =\tan^{-1}\frac{L_{I}}{N},\;\theta_{2}=\tan^{-1}\frac{L_{I}}{B+N}\\ \theta_{3}=0,\;\theta_{4}=0\\ \theta_{5}=-\tan^{-1}\frac{L_{\mathrm{II}}}{N},\;\theta_{6}=-\tan^{-1}\frac{L_{\mathrm{II}}}{B+N} \end{array}\right.$$(2)when turning left, θ<0
(2)$$\left\{ \begin{array}{l} \theta_{1}=-\tan^{-1}\frac{L_{I}}{N+B},\;\theta_{2}=\theta =-\tan^{-1}\frac{L_{I}}{N}\\ \theta_{3}=0,\;\theta_{4}=0\\ \theta_{5}=\tan^{-1}\frac{L_{\mathrm{II}}}{N+B},\;\theta_{6}=\tan^{-1}\frac{L_{\mathrm{II}}}{N} \end{array}\right.$$
where θ is the steering angle of the vehicle, which is positive when turning right or negative when turning left, and $L_{I}$ and $L_{\mathrm{II}}$ represent the distance from the front axle and rear axle to the middle shaft, respectively.


Some of the existing approaches simplified the vehicle model into a rigid body structure, resulting in lower accuracy of vehicle models. Moreover, the research field on predictive control generally simplified the car body into a spring-damped structure only with a vertical direction, which is not suitable for three-dimensional uneven roads. Considering the characteristics of the suspension and tires, we constructed a segmented spring damping system to make the model more realistic, as shown in Figure 3b.

The Euler–Lagrange equation was chosen as the dynamics equation of our vehicle through comprehensive consideration of the vehicle’s characteristics.

(3)$$\frac{d}{dt}\left(\frac{\partial (K-P)}{\partial {\dot{q}}_{i}}\right)-\frac{\partial (K-P)}{\partial q_{i}}+\frac{\partial F}{\partial {\dot{q}}_{i}}=\frac{\partial E}{\partial {\dot{q}}_{i}}$$

Assume that the coordinates of the vehicle mass center is expressed in $M(x_{m},y_{m},z_{m})$, and M(m,l,n) is the center of mass coordinates with respect to the vehicle center O(x,y,z). According to the geometric knowledge, the following equation could be deduced:(4)$$\left\{ \begin{array}{l} x_{m}=x+m+n\beta -l\gamma \\ y_{m}=y+l-n\alpha +m\gamma \\ z_{m}=z+n+l\alpha -m\beta \end{array}\right.$$
The velocity of the mass center could be deduced by taking the derivative of Equation (4):(5)$$\left\{ \begin{array}{l} {\dot{x}}_{m}=\dot{x}+n\dot{\beta }-l\dot{\gamma }\\ {\dot{y}}_{m}=\dot{y}-n\dot{\alpha }+m\dot{\gamma }\\ {\dot{z}}_{m}=\dot{z}+l\dot{\alpha }-m\dot{\beta } \end{array}\right.$$
The kinetic energy of the system K was as follows:(6)$$\begin{array}{lll} K & = & K_{l}+K_{r}\\ & = & \frac{1}{2}M\left({\dot{x}}_{m}^{2}+{\dot{y}}_{m}^{2}+{\dot{z}}_{m}^{2}\right)+\frac{1}{2}\left(\alpha^{2}J_{XX}{\dot{\alpha }}_{}^{2}+J_{YY}{\dot{\beta }}_{}^{2}+J_{ZZ}{\dot{\gamma }}_{}^{2}\right)\\ & & -\left(J_{XY}\dot{\alpha }\dot{\beta }+J_{YZ}\dot{\beta }\dot{\gamma }+J_{XZ}\dot{\alpha }\dot{\gamma }\right) \end{array}$$
where $J_{XX},J_{YY},J_{ZZ}$ are the vehicle’s moment of inertia, and $J_{XY},J_{YZ},J_{XZ}$ are the vehicle’s product of inertia.

According to the geometric relations in Figure 3a, the tangential, lateral, and radial displacement of each wheel could be calculated:(7)$$\left\{ \begin{array}{l} u_{i}=\left(x-r\beta -L_{i}\gamma \right)\cos\theta_{i}-\left[y+r\alpha +b_{i}\gamma \right]\sin\theta_{i}\\ v_{i}=\left(x-r\beta -L_{i}\gamma \right)\sin\theta_{i}+\left[y+r\alpha +b_{i}\gamma \right]\cos\theta_{i}\\ w_{i}=z-b_{i}\beta +L_{i}\alpha \end{array}\right.$$

$$b=\frac{B}{2},b_{i}={\left(-1\right)}^{i+1}b$$

The potential energy of the system could be expressed as the combination of gravitational potential energy and elastic potential energy of the vehicle. The formula was as follows:(8)$$\begin{array}{ll} P= & \frac{1}{2}(\sum_{i=1}^{6}\left[K_{ix}{\left(u_{i}-u_{i}^{\prime }\right)}^{2}\right]+\sum_{i=1}^{6}\left[K_{iy}{\left(v_{i}-v_{i}^{\prime }\right)}^{2}\right]+\sum_{i=1}^{6}\left[K_{iz}{\left(w_{i}-\delta_{0}-w_{i}^{\prime }\right)}^{2}\right])\\ & +\mathrm{Mg}\left(-x_{m}\sin\lambda \cos\varphi -y_{m}\sin\lambda \sin\varphi +z_{m}\cos\lambda \right) \end{array}$$
The energy dissipation of the system was as follows:(9)$$F=\frac{1}{2}(\sum_{i=1}^{6}\left[C_{ix}{\left({\dot{u}}_{i}-{\dot{u}}_{i}^{\prime }\right)}^{2}\right]+\sum_{i=1}^{6}\left[C_{iy}{\left({\dot{v}}_{i}-{\dot{v}}_{i}^{\prime }\right)}^{2}\right]+\sum_{i=1}^{6}\left[C_{iz}{\left({\dot{w}}_{i}-{\dot{w}}_{i}^{\prime }\right)}^{2}\right])$$
where $K_{ix},K_{iy},K_{iz}(i=1,2...6)$ and $C_{ix},C_{iy},C_{iz}(i=1,2...6)$ denote the stiffness coefficient and damping coefficient of each wheel in three directions, respectively. Further, $u_{i}^{\prime },v_{i}^{\prime },w_{i}^{\prime }(i=1,2...6)$ are the lateral, tangential, and radial displacement caused by the influence of the terrain, respectively.

Then, the work done by the wheels on the possible displacement and the force of friction on each wheel were as follows:(10)$$E=\sum_{i=1}^{6}\left[\left(P_{i}-F_{i}^{\prime }\right)v_{i}-S_{i}^{\prime }u_{i}\right]$$
(11)$$F_{i}=\mu \cdot K_{iz}\left(w_{i}-w_{i}^{\prime }\right)+C_{iz}\left({\dot{w}}_{i}-{\dot{w}}_{i}^{\prime }\right)$$
where $F_{i}^{\prime }$ and $S_{i}^{\prime }$ represent the lateral and tangential forces on the tire, respectively. $P_{i}$ denotes the traction of each wheel, which can be obtained from the controller area network (CAN) of the FTV.

Combining Equations (1)–(11), the dynamic equation of autonomous (FTV) could be deduced:
X-direction:(12)$$\begin{array}{l} M\left(\ddot{x}+n\ddot{\beta }-l\ddot{\gamma }\right)-Mg\sin\lambda \cos\varphi +\sum_{i=1}^{6}\cos\theta_{i}\cdot K_{ix}\left(u_{i}-u_{i}^{\prime }\right)+\sum_{i=1}^{6}\sin\theta_{i}\cdot K_{iy}\left(v_{i}-v_{i}^{\prime }\right)\\ +\sum_{i=1}^{6}\cos\theta_{i}\cdot C_{ix}\left({\dot{u}}_{i}-{\dot{u}}_{i}^{\prime }\right)+\sum_{i=1}^{6}\sin\theta_{i}\cdot C_{iy}\left({\dot{v}}_{i}-{\dot{v}}_{i}^{\prime }\right)=0 \end{array}$$
Y-direction:(13)$$\begin{array}{l} M\left(\ddot{y}-n\ddot{\alpha }+m\ddot{\gamma }\right)-Mg\sin\lambda \sin\varphi -\sum_{i=1}^{6}\sin\theta_{i}\cdot K_{ix}\left(u_{i}-u_{i}^{\prime }\right)+\sum_{i=1}^{6}\cos\theta_{i}\cdot K_{iy}\left(v_{i}-v_{i}^{\prime }\right)\\ -\sum_{i=1}^{6}\sin\theta_{i}\cdot C_{ix}\left({\dot{u}}_{i}-{\dot{u}}_{i}^{\prime }\right)+\sum_{i=1}^{6}\cos\theta_{i}\cdot C_{iy}\left({\dot{v}}_{i}-{\dot{v}}_{i}^{\prime }\right)=0 \end{array}$$
Z-direction:(14)$$M(\ddot{z}+l\ddot{\alpha }-m\ddot{\beta })+Mgcos\lambda +\sum_{i=1}^{6}K_{iz}\left(w_{i}-\delta_{0}-w_{i}^{\prime }\right)+\sum_{i=1}^{6}C_{iz}\left({\dot{w}}_{i}-{\dot{w}}_{i}^{\prime }\right)=0$$
α-direction:(15)$$\begin{array}{l} -Mn\ddot{y}+Ml\ddot{z}+\left[J_{XX}+M\left(n^{2}+l^{2}\right)\right]\ddot{\alpha }-\left(Mml+J_{XY}\right)\ddot{\beta }-\left(Mmn+J_{XZ}\right)\ddot{\gamma }\\ +Mg\left(n\sin\lambda \sin\varphi +l\cos\lambda \right)-\sum_{i=1}^{6}\left[r\cdot \sin\theta_{i}\cdot K_{ix}\left(u_{i}-u_{i}^{\prime }\right)+r\cdot \cos\theta_{i}\cdot K_{iy}\left(v_{i}-v_{i}^{\prime }\right)+L_{i}K_{iz}\left(w_{i}-\delta_{0}-w_{i}^{\prime }\right)\right]\\ +\sum_{i=1}^{6}\left[-r\cdot \sin\theta_{i}\cdot C_{ix}\left({\dot{u}}_{i}-{\dot{u}}_{i}^{\prime }\right)+r\cdot \cos\theta_{i}\cdot C_{iy}\left({\dot{v}}_{i}-{\dot{v}}_{i}^{\prime }\right)+L_{i}\cdot C_{iz}\left({\dot{w}}_{i}-{\dot{w}}_{i}^{\prime }\right)\right]=0 \end{array}$$
β-direction:(16)$$\begin{array}{l} Mn\ddot{x}-Mm\ddot{z}-\left(J_{XY}+Mml\right)\ddot{\alpha }+\left[J_{YY}+M\left(m^{2}+n^{2}\right)\right]\ddot{\beta }-\left(J_{YZ}+Mnl\right)\ddot{\gamma }\\ -Mg\left(n\sin\lambda \cos\varphi +m\cos\lambda \right)-\sum_{i=1}^{6}\left[r\cdot \cos\theta_{i}\cdot K_{ix}\left(u_{i}-u_{i}^{\prime }\right)+r\cdot \sin\theta_{i}\cdot K_{iy}\left(v_{i}-v_{i}^{\prime }\right)+b_{i}K_{iz}\left(w_{i}-\delta_{0}-w_{i}^{\prime }\right)\right]\\ +\sum_{i=1}^{6}\left[r\cdot \cos\theta_{i}\cdot C_{ix}\left({\dot{u}}_{i}-{\dot{u}}_{i}^{\prime }\right)-r\cdot \sin\theta_{i}\cdot C_{iy}\left({\dot{v}}_{i}-{\dot{v}}_{i}^{\prime }\right)-b_{i}\cdot C_{iz}\left({\dot{w}}_{i}-{\dot{w}}_{i}^{\prime }\right)\right]=0 \end{array}$$
γ-direction:(17)$$\begin{array}{l} -Ml\ddot{x}+Mm\ddot{y}-\left(J_{XZ}+Mmn\right)\ddot{\alpha }-\left(J_{YZ}+Mnl\right)\ddot{\beta }+\left[J_{ZZ}+M\left(l^{2}+m^{2}\right)\right]\ddot{\gamma }\\ +Mg\left(l\sin\lambda \cos\varphi -m\sin\lambda \sin\varphi \right)+\sum_{i=1}^{6}\left(-L_{i}\cos\theta_{i}-b_{i}\sin\theta_{i}\right)\cdot \left[K_{ix}\cdot \left(u_{i}-u_{i}^{\prime }\right)+C_{ix}\cdot \left({\dot{u}}_{i}-{\dot{u}}_{i}^{\prime }\right)\right]\\ +\sum_{i=1}^{6}\left(-L_{i}\sin\theta_{i}+b_{i}\cos\theta_{i}\right)\cdot \left[K_{iy}\cdot \left(v_{i}-v_{i}^{\prime }\right)+C_{iy}\cdot \left({\dot{v}}_{i}-{\dot{v}}_{i}^{\prime }\right)\right]=0 \end{array}$$
$u_{i}$-direction:(18)$$K_{ix}\cdot \left(u_{i}-u_{i}^{\prime }\right)+C_{ix}\cdot \left({\dot{u}}_{i}-{\dot{u}}_{i}^{\prime }\right)=S_{i}(i=1,2...6)$$
$v_{i}$-direction:(19)$$K_{iy}\cdot \left(v_{i}-v_{i}^{\prime }\right)+C_{iy}\cdot \left({\dot{v}}_{i}-{\dot{v}}_{i}^{\prime }\right)=P_{i}-F_{i}(i=1,2...6)$$
The matrix form of the above equation was expressed as follows:
(20)$$\left[\begin{array}{cc} \left[M_{6\times 6}\right] & {\left[0\right]}_{6\times 12}\\ {\left[0\right]}_{12\times 6} & {\left[0\right]}_{12\times 12} \end{array}\right]\left\{\begin{array}{c} {\ddot{q}}_{6}\\ {\ddot{q}}_{12} \end{array}\right\}+\left[\begin{array}{cc} \left[C_{6\times 6}\right] & \left[C_{6\times 12}\right]\\ \left[C_{12\times 6}\right] & \left[C_{12\times 12}\right] \end{array}\right]\left\{\begin{array}{c} {\dot{q}}_{6}\\ {\dot{q}}_{12} \end{array}\right\}+\left[\begin{array}{cc} \left[K_{6\times 6}\right] & \left[K_{6\times 12}\right]\\ \left[K_{12\times 6}\right] & \left[K_{12\times 12}\right] \end{array}\right]\left\{\begin{array}{c} q_{6}\\ q_{12} \end{array}\right\}=\left\{\begin{array}{c} F_{6}\\ F_{12} \end{array}\right\}$$
$$\left[M_{6\times 6}\right]=\left[\begin{array}{cccccc} M & 0 & 0 & 0 & M\cdot n & -M\cdot l\\ 0 & M & 0 & -M\cdot n & 0 & M\cdot m\\ 0 & 0 & M & M\cdot l & -M\cdot m & 0\\ 0 & -M\cdot n & M\cdot l & J_{XX}+M\cdot (n^{2}+l^{2}) & -(J_{XY}+M\cdot m\cdot l) & -(J_{ZX}+M\cdot m\cdot n)\\ M\cdot n & 0 & -M\cdot m & -(J_{XY}+M\cdot m\cdot l) & J_{YY}+M\cdot (n^{2}+m^{2}) & -(J_{YZ}+M\cdot l\cdot n)\\ -M\cdot l & M\cdot m & 0 & -(J_{ZX}+M\cdot m\cdot n) & -(J_{YZ}+M\cdot l\cdot n) & J_{ZZ}+M\cdot (l^{2}+m^{2}) \end{array}\right]$$
$$\left\{q_{6}\right\}=\left\{x\;y\;z\;\alpha \;\beta \;\gamma \right\},\left\{q_{12}\right\}=\left\{u_{1}^{\prime }...u_{6}^{\prime }\;v_{1}^{\prime }...v_{6}^{\prime }\right\}$$
where $\left[C_{6\times 6}\right]$, $\left[C_{6\times 12}\right]$, $\left[C_{12\times 12}\right]$ consists of the coefficient of $\left\{{\dot{q}}_{6}\right\}$ and $\left\{{\dot{q}}_{12}\right\}$ in Equations (12)–(19); $\left[K_{6\times 6}\right]$, $\left[K_{6\times 12}\right]$, $\left[K_{12\times 12}\right]$ consists of the coefficient of and in Equations (12)–(19); $F_{6}$ and $F_{12}$ consist of the generalized force in Equations (12)–(19). All of the above parameters are known.

### 3.2. Kalman Filter Algorithm

The dynamic model derived in the previous chapter could be used to calculate the pose when the vehicle was on an uneven road. However, the accuracy of the predicted pose was low, owing to the errors caused by IMU and the accuracy of the dynamic model. Several methods have been proposed to compensate for these errors. An effective method is the Kalman filter technique, which was proposed by Kalman in 1960 [28]. In this paper, the KF algorithm was used to compensate for the errors caused by a dynamic model and IMU for improving the accuracy of pose predicted.

The KF algorithm mainly includes two steps: predict and update. In the predict step, the state of the system is predicted with the following two equations: (21)$${\hat{X}}_{k}=A_{k}{\hat{X}}_{k-1}+B_{k}{\vec{u}}_{k}+w_{k}$$
(22)$${\hat{P}}_{k}=A_{k}{\hat{P}}_{k-1}A_{k}{}^{T}+Q_{k-1}$$
where ${\hat{X}}_{k}$ and ${\hat{P}}_{k}$ represent the system predicted vector and predicted covariance matrix at time $t_{k}$, respectively; ${\hat{X}}_{k-1}$ and ${\hat{P}}_{k-1}$ represent the system condition vector and system covariance matrix at time $t_{k-1}$, respectively; $w_{k}$ and $Q_{k-1}$ represent system error and corresponding covariance matrix at time $t_{k-1}$, respectively.

The above equations are linear system equations, which could not be used in the dynamic model. Thus, we needed to convert Equation (20) into a linear system equation to match the Kalman filter algorithm.

Firstly, Equation (20) could be rewritten as follows: (23)$$\left\{{\ddot{q}}_{18}\right\}={\left[M_{18\times 18}\right]}^{-1}\left\{F_{18}\right\}-{\left[M_{18\times 18}\right]}^{-1}\left[C_{18\times 18}\right]\left\{{\dot{q}}_{18}\right\}-{\left[M_{18\times 18}\right]}^{-1}\left[K_{18\times 18}\right]\left\{q_{18}\right\}$$

Then, a vector with 24 state variables was used to convert Equation (23) into a 1-order differential equation, and Equation (23) could be rewritten as follows:(24)$$\left\{\dot{X}\right\}=\left[E\right]\left\{X\right\}+\left\{F^{*}\right\}$$

$$\left\{X\right\}=\left\{\begin{array}{c} q_{18}\\ {\dot{q}}_{6} \end{array}\right\}=\left\{x\;y\;z\;\alpha \;\beta \;\gamma \;u_{1}^{\prime }\ldots u_{6}^{\prime }\;v_{1}^{\prime }\ldots v_{6}^{\prime }\;\dot{x}\;\dot{y}\;\dot{z}\;\dot{\alpha }\;\dot{\beta }\;\dot{\gamma }\right\}$$

$$\left[E\right]=\left[\begin{array}{ccc} {\left[0\right]}_{6\times 6} & {\left[0\right]}_{6\times 12} & {\left[I\right]}_{6\times 6}\\ -{\left[C_{12\times 12}\right]}^{-1}\left[K_{12\times 6}\right] & -{\left[C_{12\times 12}\right]}^{-1}\left[K_{12\times 12}\right] & -{\left[C_{12\times 12}\right]}^{-1}\left[C_{12\times 6}\right]\\ \left[T_{6\times 12}\right]\left[K_{12\times 6}\right]-{\left[M_{6\times 6}\right]}^{-1}\left[K_{6\times 6}\right] & \left[T_{6\times 12}\right]\left[K_{12\times 12}\right]-{\left[M_{6\times 6}\right]}^{-1}\left[K_{6\times 12}\right] & \left[T_{6\times 12}\right]\left[C_{12\times 6}\right]-{\left[M_{6\times 6}\right]}^{-1}\left[C_{6\times 6}\right] \end{array}\right]$$

$$\left\{F^{*}\right\}=\left\{\begin{array}{c} {\left\{0\right\}}_{6}\\ {\left[C_{12\times 12}\right]}^{-1}\left\{F_{12}\right\}\\ {\left[M_{6\times 6}\right]}^{-1}\left\{F_{6}\right\}-\left[T_{6\times 12}\right]\left\{F_{12}\right\} \end{array}\right\}$$

$$\left[T_{6\times 12}\right]={\left[M_{6\times 6}\right]}^{-1}\left[C_{6\times 12}\right]{\left[C_{12\times 12}\right]}^{-1}$$

Secondly, we used a 4-order Runge–Kutta equation to solve Equation (24) for combining with the Kalman filter algorithm:(25)$$\left\{ \begin{array}{l} {\left\{X\right\}}_{k}={\left\{X\right\}}_{k-1}+\frac{1}{6}(K_{1}+2K_{2}+2K_{3}+K_{4})\\ K_{1}=\Delta T\cdot \Phi ({\left\{X\right\}}_{k-1})\\ K_{2}=\Delta T\cdot \Phi ({\left\{X\right\}}_{k-1}+\frac{1}{2}K_{1})\\ K_{3}=\Delta T\cdot \Phi ({\left\{X\right\}}_{k-1}+\frac{1}{2}K_{2})\\ K_{4}=\Delta T\cdot \Phi ({\left\{X\right\}}_{k-1}+\frac{1}{2}K_{3}) \end{array}\right.$$

Finally, Equations (21) and (22) were rewritten as follows:(26)$$X_{k}=A_{k}X_{k-1}+F_{k}+w_{k}$$
(27)$$P_{k}=A_{k}P_{k-1}A_{k}{}^{T}+Q_{k-1}$$
(28)$$A_{k}=\phi {\left([E\right]}_{k-1})$$
(29)$$B_{k}=\psi ({\left\{F^{*}\right\}}_{k-1})$$
where $X_{k-1}$ and $P_{k-1}$ represent state vector of the vehicle consisting of 24 variables and corresponding covariance matrix at time $t_{k-1}$, respectively; $X_{k}$ and $P_{k}$ represent the predicted state vector and predicted covariance matrix at time $t_{k}$, respectively; $A_{k}$ is a matrix composed of mass, stiffness coefficient, and damping coefficient of the vehicle at time $t_{k-1}$; $F_{k}$ is a matrix composed of the vehicle’s generalized force at time $t_{k-1}$; $A_{k}$ and $B_{k}$ can be calculated through Equation (25); $Q_{k}$ is the system noise covariance at time $t_{k-1}$.

The state vector of the vehicle measured by sensors could be calculated with the following two equations: (30)$$Z_{k}=H_{k}{\bar{X}}_{k}+v_{k}$$
(31)$$S_{k}=H_{k}{\hat{P}}_{k-1}H_{k}{}^{T}+R_{k}$$
where ${\bar{X}}_{k}$ is a state vector measured by sensors; $v_{k}$ and $R_{k}$ represent the observation noise and corresponding covariance matrix of sensors, respectively.

In the update step, the state and covariance estimates of the heavy-duty vehicle were corrected by the following equations:(32)$$K_{k}={\hat{P}}_{k}H_{k}{}^{T}{\left[H_{k}{\hat{P}}_{k}H_{k}{}^{T}+R_{k}\right]}^{-1}$$

(33)$$P_{k}=[I+H_{k}K_{k}]{\hat{P}}_{k}$$

(34)$$X_{k}={\hat{X}}_{k}+K_{k}[Z_{k}-H_{k}{\hat{X}}_{k}]$$

The flow chart of the Kalman filter is shown in Figure 4:

### 3.3. Active Suspension System Control

The active suspension system can control the vehicle vibration and pose by changing the height, shape, and damping of the suspension system, so as to improve the performance of the vehicle’s operating stability and ride comfort. The automobile’s active suspension system can be divided into three categories according to the control type: hydraulic control suspension system, air suspension system, and electromagnetic induction suspension system. Our car adopts a hydraulic suspension system to support such a heavy body of FTV. Most of the methods on the control of suspension are based on the IMU or other three-dimensional sensors on the car to measure the state of the car, and then according to the state, to adjust the suspension to maintain the stability. However, this method has some defects. On the one hand, IMU does not output data in continuous time, which leads to the inaccuracy of suspension control. On the other hand, the INS only measures the current state of the vehicle, which results in a delay in suspension adjustment. Our proposed pose prediction could not only solve the discontinuity of IMU but also solve the delay of IMU.

Figure 5 shows the active suspension system of our FTV. According to the geometric relationship, the kinematic equation of FTV could be obtained:(35)$$\left\{ \begin{array}{l} Z_{1}=Z-L_{f}\sin\theta +\frac{1}{2}a\cos\theta \sin\varphi \\ Z_{2}=Z-L_{f}\sin\theta -\frac{1}{2}a\cos\theta \sin\varphi \\ Z_{3}=Z+L_{r}\sin\theta +\frac{1}{2}a\cos\theta \sin\varphi \\ Z_{4}=Z+L_{r}\sin\theta -\frac{1}{2}a\cos\theta \sin\varphi \\ Z_{5}=Z+\frac{1}{2}a\sin\varphi \\ Z_{6}=Z-\frac{1}{2}a\sin\varphi \end{array}\right.$$
where θ, φ, and Z could be predicted through our dynamic model. Then, we could get the z-direction displacement of the body at the connection with the hydraulic cylinder. Finally, we controlled the hydraulic cylinder to maintain the stability and safety of the FTV.

## 4. Discussion

We used four experiments to verify the effectiveness and feasibility of our method. The first experiment was to confirm that our pose prediction using a Kalman filter was more accurate and stable than using the dynamic model alone. The second experiment was to verify the accuracy of our proposed method over a while compared with the real vehicle pose. The third experiment was to verify the effectiveness of our active suspension system control method. The last experiment was to verify the stability of our method in more situations for a long period.

The experiment was first carried out on the test site with two kinds of obstacles, as shown in Figure 6a. We used the nearest neighbor interpolation (NNI) algorithm to process cloud data points obtained from 3D sensors, as shown in Figure 6b.

The vehicle’s pose at time $t_{k+1}$ was predicted by inputting information about the vehicle and the road at time $t_{k}$ to the dynamic model and compared with the data of GPS/INS at time $t_{k+1}$. The position errors relative to GPS/INS data are shown in Figure 7. The blue dotted line represents position error only predicted by the dynamic model. As the curve shows, using the dynamic model to predict the pose produced a large error due to its iterative algorithm and the accuracy of sensors mounted on the vehicle. The red solid line represents position error predicted by the dynamic model using a Kalman filter algorithm. The curve demonstrated that the combination of a dynamic model and Kalman filter algorithm could effectively eliminate the error, which could ensure accurate positioning. The attitude angle errors are shown in Figure 8. For the same reason, using a Kalman filter algorithm could get more accurate positioning and attitude angle information.

For our autonomous FTV, active suspension system control and advanced risk analysis were implemented to maintain stability and safety on uneven terrain. To achieve the above method, we needed to accurately predict the attitude information of the vehicle in the next period of time. The second experiment was carried out to verify the effect of our method. We made the vehicle pass an obstacle directly at a constant slow speed, and, at the same time, recorded the vehicle pose information and vehicle input information in this period. Then, we imported the same vehicle input information into the dynamic model to get the prediction information. The result is shown in Figure 9—the purple line and the red line show the attitude angle information of the vehicle and the predicted attitude angle information, respectively. The result showed that the predicted pose had high accuracy. It means that we could calculate how much speed and traction were needed to pass some obstacles, as there was a risk of rollover when passing obstacles at a certain speed in advance.

The data after adjusting the suspension of the vehicle are shown in Figure 10—the purple line shows the attitude angle information when passing through an obstacle, and the red line shows the attitude angle information after the vehicle continuously adjusts the suspension when passing through the same obstacle at the same speed. The result showed that the attitude was maintained within 5 degrees by continuous suspension adjustment, which means that the stability of the vehicle could be well guaranteed by our proposed method.

The last experiment was carried out to further verify the feasibility of our method and its stability in complex environments. The experiment was carried out in a field around our university, with various obstacles on the road. As shown in Figure 11b, the black line shows the real trajectory of the FTV, and the blue line and red line represent the trajectories predicted by the dynamic model with and without Kalman filter, respectively. The experiment result showed that the method, combining a Kalman filter with a dynamic model, had higher accuracy and stability over a long period. The three points where the car passed through the obstacle were A, B, and C. Figure 12 shows the change of attitude angle during driving. The vehicle attitude angle maintained within 5 degrees during the whole driving process. The result showed that the vehicle could maintain stability for a long time through suspension system control.

## 5. Conclusions

This paper presented a method of pose prediction of autonomous FTV on uneven roads, which could be used as a reference index for adjusting active suspension and planning path. The first was the way to extract road real-time elevation information through GPS/INS and 3D laser scanners. The second was the description of a method that established the vehicle’s dynamic model and imported the elevation information extracted from the previous step into it for pose estimation. Finally, the dynamic model was combined with a KF to obtain a more accurate pose prediction. Real experiments results demonstrated that the safety and stability of vehicles driving across complex uneven terrain could be ensured by using our method. There were also some defects in the whole study. For example, the accuracy of the vehicle’s dynamic model was reduced when adjusting the active suspension system. In the future, we would consider how to improve the accuracy of a dynamic model of the vehicle and adjust the suspension system in more forms to maintain the vehicle’s stability. Furthermore, we need to do in-depth research on the path planning across complex uneven terrain to make the vehicle truly unmanned.

## Figures and Tables

**Figure 1 sensors-19-05120-f001:**
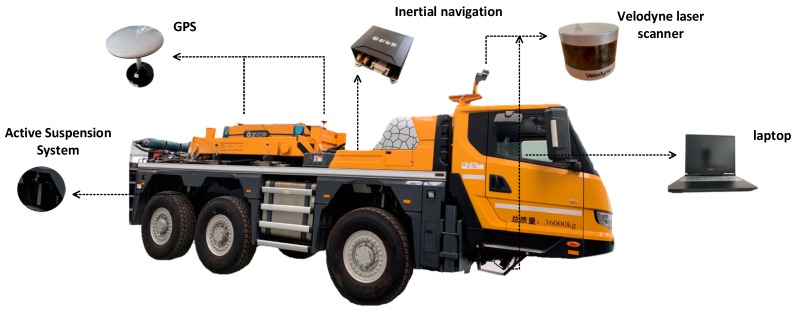
Vehicle configuration.

**Figure 2 sensors-19-05120-f002:**
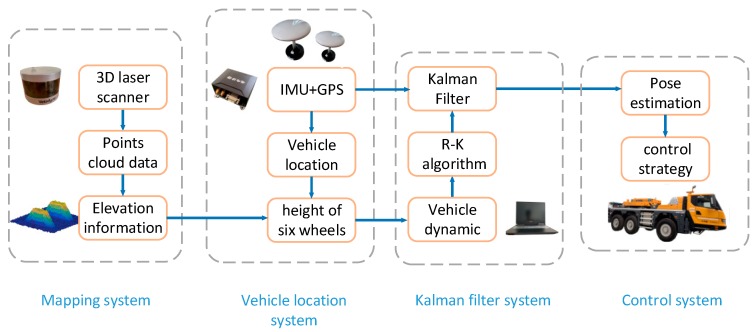
System structure. IMU, inertial measurement unit; R–K, Runge–Kutta.

**Figure 3 sensors-19-05120-f003:**
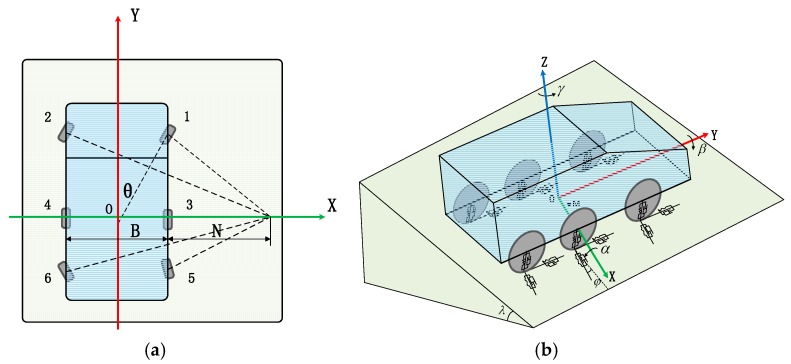
Autonomous heavy-duty vehicle models. (**a**) Top view of the vehicle. (**b**) A simplified model of the vehicle on a slope.

**Figure 4 sensors-19-05120-f004:**
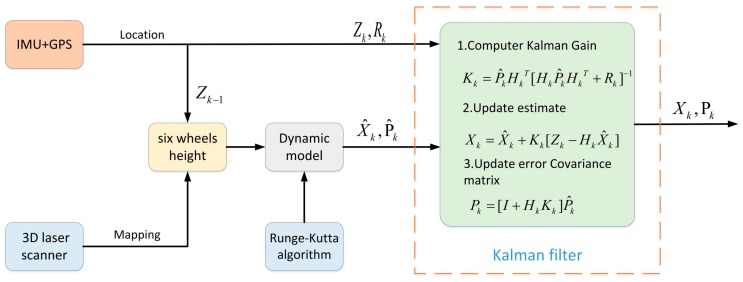
Kalman filter algorithm.

**Figure 5 sensors-19-05120-f005:**
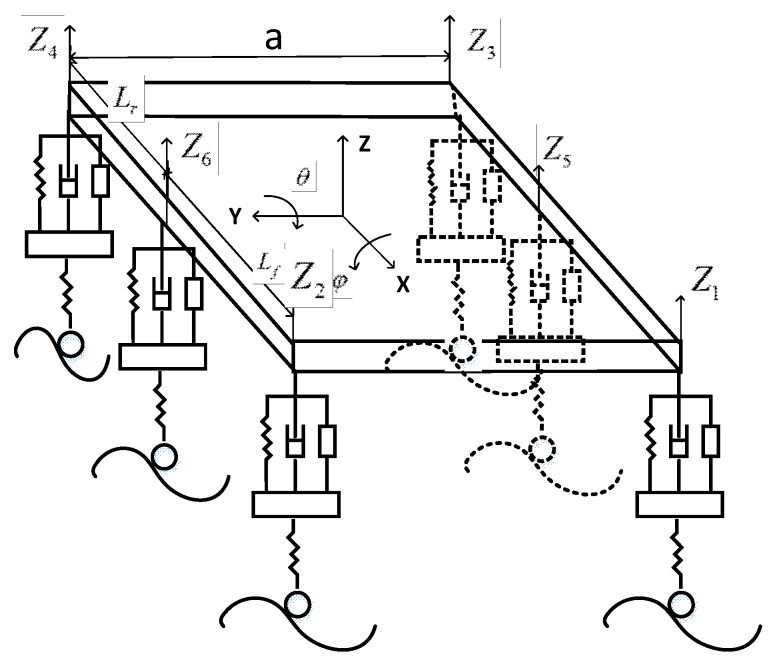
Model of the active suspension system.

**Figure 6 sensors-19-05120-f006:**
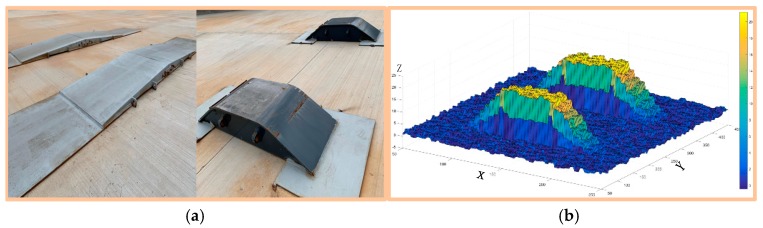
Two kinds of obstacles in the test site. (**a**) Real obstacles in the test site. (**b**) Points cloud data of an obstacle processed by the nearest neighbor interpolation (NNI) algorithm.

**Figure 7 sensors-19-05120-f007:**
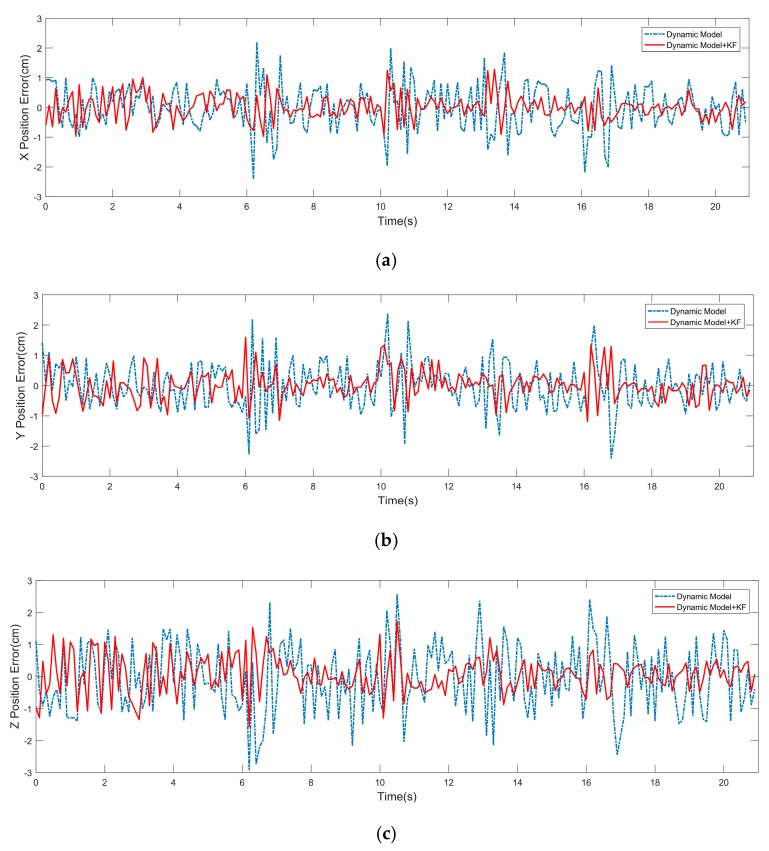
Positioning error with and without Kalman filter algorithm. (**a**) X position error. (**b**) Y position error. (**c**) Z position error.

**Figure 8 sensors-19-05120-f008:**
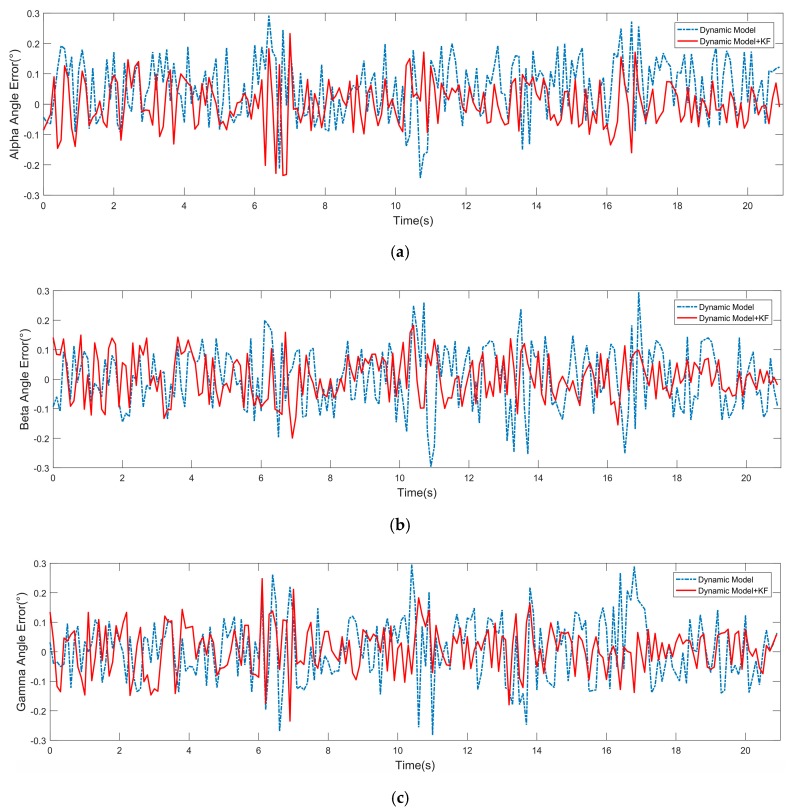
Angle errors with and without the Kalman filter algorithm. (**a**) Alpha angle error. (**b**) Beta angle error. (**c**) Gamma angle error.

**Figure 9 sensors-19-05120-f009:**
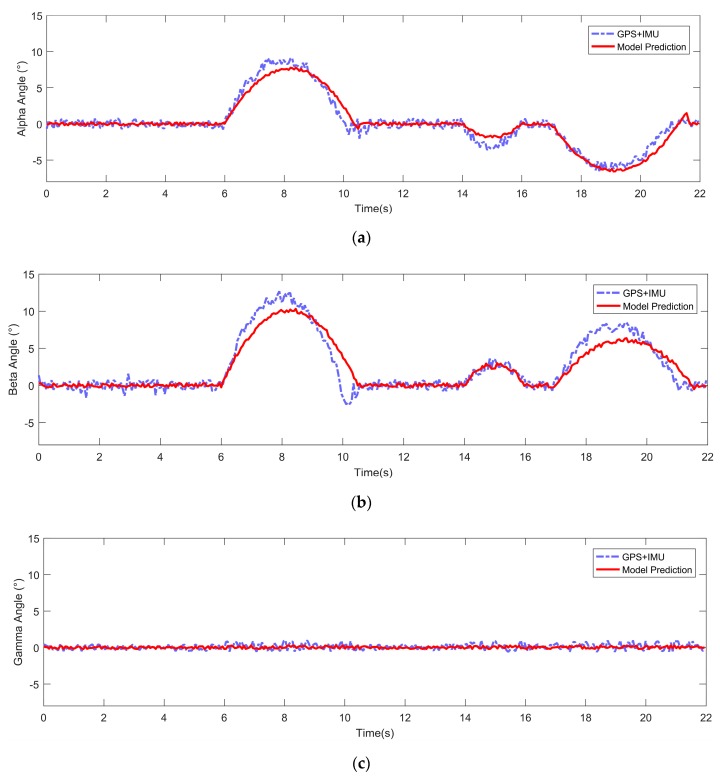
Prediction data of the vehicle’s attitude angle. (**a**) Alpha angle. (**b**) Beta angle. (**c**) Gamma angle.

**Figure 10 sensors-19-05120-f010:**
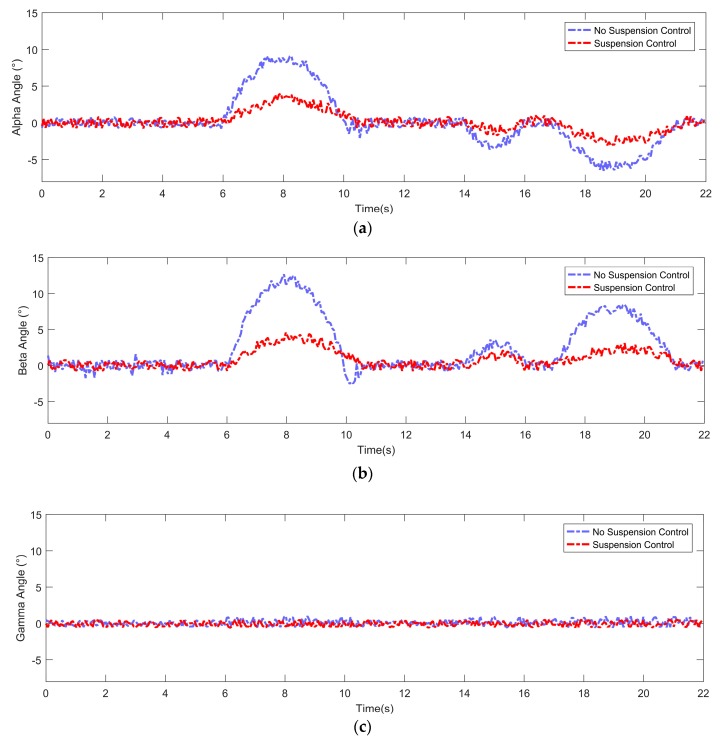
Attitude angle information after adjusting the active suspension system. (**a**) Alpha angle. (**b**) Beta angle. (**c**) Gamma angle.

**Figure 11 sensors-19-05120-f011:**
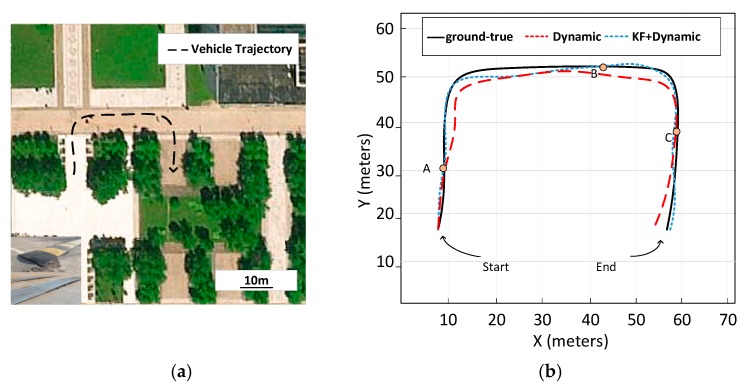
Test sites around our university and vehicle trails. (**a**) Top view of test sites. (**b**) The trajectory curves of the full tracked vehicle (FTV).

**Figure 12 sensors-19-05120-f012:**
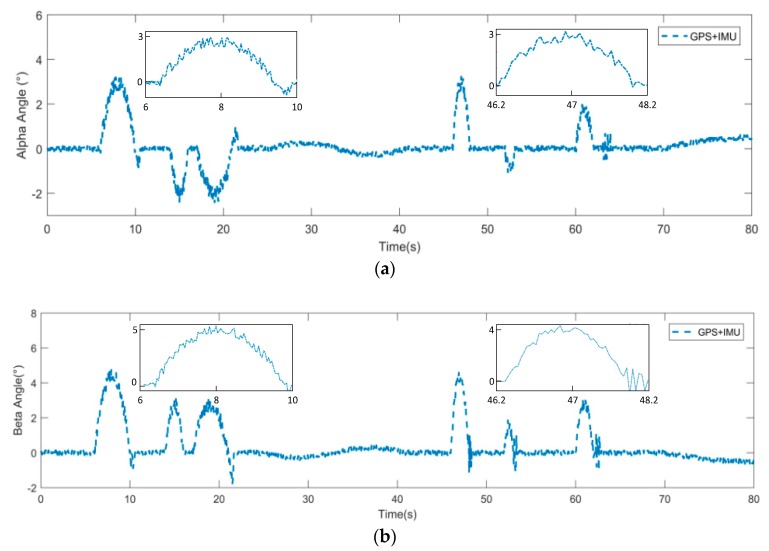
Data on the vehicle’s attitude angle from the whole process of vehicle movement. (**a**) Alpha angle. (**b**) Beta angle.

**Table 1 sensors-19-05120-t001:** Performance index of 3D-sensor.

Performance Index	Parameter
Scanning frequency	5–20 Hz
Measuring range	0–100 mm
Horizontal angular resolution	0.1°–0.4°
Vertical angular resolution	2°
Precision of distance measurement	±3 cm
Number of laser lines	16 L

**Table 2 sensors-19-05120-t002:** Performance index of GPS/INS.

Performance Index	Parameter
Data out frequency	100 Hz
Precision of heading	±0.3°
Precision of horizontal attitude	±0.3°
Precision of horizontal position	±2 cm
Precision of Altitude position	±3 cm

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
