# Peer review of "Pose Prediction of Autonomous Full Tracked Vehicle Based on 3D Sensor"

_sensors, 2019, doi:10.3390/s19235120_

Round 1
Reviewer 1 Report
This manuscript addresses an autonomous mobile pose prediction system. It uses two Velodyne 3D sensors and IMU as the acquisition system and a Kalman Filter methodology to perform the sensor fusion. It is an actual and interesting paper.
Some typos that should be corrected:
vehicle(FTV) -> vehicle (FTV)
points cloud data of terrain -> cloud data points of terrain (several times)
location -> Location
Equation 2: Theta1 to 6: please describe them (wheels angles? Which ones?)
Eq.3 also uses some variables not addressed on text.
Fig.4: please put in the same line “Figure .” -> 4 , Kalmenn,,,,,
Fig5 b) please put x, y, z into the axes.
Security should also be addressed since it is a very import topic in such vehicle, due to its weight and size. (how is it assured?)
Results should be improved with more situations.
Reviewer 2 Report
This paper is well written. Below are some comments:
How to get (20) from (1)-(19)? It cannot be too simple. Why is the equation (20) linear rather than nonlinear? It may be wrong to claim that "The equation (24) was converted into the 4-order Runge-Kutta equation". We can say "use the 4-order Runge-Kutta algorithm to solve the equation (24). "when turn right", "when turn left", here "turn" should be "turning". By the way, some "Where" should be "where".
Reviewer 3 Report
The paper deals with the design of a navigation system for autonomous vehicle. In the present form the paper presents many lacks. Starting from the introdcution authors should extend the literature review in or der to provide to the reader a wider picture on the research domain. Some more paper should be added as for instance the following ones:
Grivon, D. et al. (2013). Development of an innovative low-cost MARG sensors alignment and distortion compensation methodology for 3D scanning applications. Robotics and Autonomous Systems, 61(12), 1710-1716.
Also some more issues regarding the methodological section should be added in order to better explain the proposed system from a global point of view together with some more details regarding the paramters involved in the different elements.
Some more lines should be provided for what concerns the experimental validation, expecially on the experimental setting, that represents a key elements for really understing the results obtained
Round 2
Reviewer 3 Report
Authors have improved the scientific level of the paper I support the paper publication